# Inequities in COVID-19 vaccine and booster coverage across Massachusetts ZIP codes after the emergence of Omicron: A population-based cross-sectional study

Jacob Bor[1,2]*, Sabrina A. Assoumou[3,4], Kevin Lane[5], Yareliz Diaz[6], Bisola O. Ojikutu[7], Julia Raifman[6], Jonathan I. Levy[5]

1 Department of Global Health, Boston University School of Public Health, Boston, Massachusetts, United States of America, 2 Department of Epidemiology, Boston University School of Public Health, Boston, Massachusetts, United States of America, 3 Section of Infectious Diseases, Boston University School of Medicine, Boston, Massachusetts, United States of America, 4 Boston Medical Center, Boston, Massachusetts, United States of America, 5 Department of Environmental Health, Boston University School of Public Health, Boston, Massachusetts, United States of America, 6 Department of Health Law, Policy, and Management, Boston University School of Public Health, Boston, Massachusetts, United States of America, 7 Boston Public Health Commission, City of Boston, Boston, Massachusetts, United States of America

* jbor@bu.edu

## Abstract

### Background

Inequities in Coronavirus Disease 2019 (COVID-19) vaccine and booster coverage may contribute to future disparities in morbidity and mortality within and between Massachusetts (MA) communities.

### Methods and findings

We conducted a population-based cross-sectional study of primary series vaccination and booster coverage 18 months into the general population vaccine rollout. We obtained public-use data on residents vaccinated and boosted by ZIP code (and by age group: 5 to 19, 20 to 39, 40 to 64, 65+) from MA Department of Public Health, as of October 10, 2022. We constructed population denominators for postal ZIP codes by aggregating census tract population estimates from the 2015–2019 American Community Survey. We excluded nonresidential ZIP codes and the smallest ZIP codes containing 1% of the state's population. We mapped variation in ZIP code-level primary series vaccine and booster coverage and used regression models to evaluate the association of these measures with ZIP code-level socioeconomic and demographic characteristics. Because age is strongly associated with COVID-19 severity and vaccine access/uptake, we assessed whether observed socioeconomic and racial/ethnic inequities persisted after adjusting for age composition and plotted age-specific vaccine and booster coverage by deciles of ZIP code characteristics.

We analyzed data on 418 ZIP codes. We observed wide geographic variation in primary series vaccination and booster rates, with marked inequities by ZIP code-level education, median household income, essential worker share, and racial/ethnic composition. In age-

**Data Availability Statement:** Data are publicly available for download from the Massachusetts Department of Public Health (https://www.mass.gov/info-details/massachusetts-covid-19-vaccination-data-and-updates) and US Census

Bureau (https://www.census.gov/programs-surveys/acs/data/data-tables.html).

**Funding:** The author(s) received no specific funding for this work.

**Competing interests:** The authors have declared that no competing interests exist.

**Abbreviations:** ACS, American Community Survey; CI, confidence interval; COVID-19, Coronavirus Disease 2019; DPH, Department of Public Health; MA, Massachusetts; MIIS, Massachusetts Immunization Information System; VEI, Vaccine Equity Initiative; ZCTA, ZIP code tabulation area.

stratified analyses, primary series vaccine coverage was very high among the elderly. However, we found large inequities in vaccination rates among younger adults and children, and very large inequities in booster rates for all age groups. In multivariable regression models, each 10 percentage point increase in "percent college educated" was associated with a 5.1 (95% confidence interval (CI) 3.9 to 6.3, $p < 0.001$) percentage point increase in primary series vaccine coverage and a 5.4 (95% CI 4.5 to 6.4, $p < 0.001$) percentage point increase in booster coverage. Although ZIP codes with higher "percent Black/Latino/Indigenous" and higher "percent essential workers" had lower vaccine coverage (−0.8, 95% CI −1.3 to −0.3, $p < 0.01$; −5.5, 95% CI −7.3 to −3.8, $p < 0.001$), these associations became strongly positive after adjusting for age and education (1.9, 95% CI 1.0 to 2.8, $p < 0.001$; 4.8, 95% CI 2.6 to 7.1, $p < 0.001$), consistent with high demand for vaccines among Black/Latino/Indigenous and essential worker populations within age and education groups. Strong positive associations between "median household income" and vaccination were attenuated after adjusting for age. Limitations of the study include imprecision of the estimated population denominators, lack of individual-level sociodemographic data, and potential for residential ZIP code misreporting in vaccination data.

## Conclusions

Eighteen months into MA's general population vaccine rollout, there remained large inequities in COVID-19 primary series vaccine and booster coverage across MA ZIP codes, particularly among younger age groups. Disparities in vaccination coverage by racial/ethnic composition were statistically explained by differences in age and education levels, which may mediate the effects of structural racism on vaccine uptake. Efforts to increase booster coverage are needed to limit future socioeconomic and racial/ethnic disparities in COVID-19 morbidity and mortality.

## Author summary

### Why was this study done?

- Vaccination, including the booster shot, is a critical line of defense against severe disease due to Coronavirus Disease 2019 (COVID-19) infection.

- We sought to understand the geography of vaccine and booster coverage across Massachusetts (MA) ZIP codes and to assess coverage inequities 18 months into the MA's general population vaccine rollout.

### What did the researchers do and find?

- Data on numbers vaccinated and boosted from MA Department of Public Health were combined with ZIP code denominators constructed de novo from US Census data.

- As of October 2022, there were large differences in primary series vaccine and booster coverage across MA ZIP codes. The share of children vaccinated ranged from under

40% to over 90% across ZIP codes. The share of elderly adults boosted ranged from under 60% to 100%.

- Primary series vaccine and booster coverage increased with ZIP code-level income and education and fell with percent Black/Latino/Indigenous and percent essential workers. Education levels were the strongest predictor of vaccine and booster uptake.

- After adjusting for age and education levels, vaccine and booster uptake was higher in ZIP codes with many Black/Latino/Indigenous residents or essential workers. Access, not "hesitancy," may drive persistent vaccination gaps in these communities.

### What do these findings mean?

- Inequities in vaccine and booster coverage may lead to future inequities in morbidity, mortality, and economic losses due to COVID-19.

## Introduction

Vaccination against Coronavirus Disease 2019 (COVID-19)—including recommended booster shots—is a critical line of defense against severe illness, hospitalization, and death. In Fall 2022, the United States had ended most public health measures amidst a continued high number of daily deaths and the potential for future surges due to holiday travel, waning immunity, or emergence of new variants. Communities with low vaccination rates are particularly vulnerable to future COVID-19 infection [1,2]. Vaccine uptake in Massachusetts (MA) is high relative to the national average. While the Winter 2021–2022 wave resulted in few excess deaths in MA, there was a rise in hospitalizations [3]. Still, as of October 2022, there were an appreciable number of eligible individuals who remained unvaccinated and a greater number who had not received boosters.

Nationally, vaccination rates are lower for less educated and lower-income people, and vaccine uptake has been slower in Black and Latino populations [4,5]. These populations, as well as the overlapping population of essential workers, have been at elevated risk for COVID exposure, infection, morbidity, and mortality throughout the pandemic [6–12] Inequities in vaccine coverage could further exacerbate disparities in health outcomes.

"Vaccine hesitancy" has dominated media coverage on racial disparities, often without mention of the historical roots of medical mistrust [13]. However, other factors associated with structural racism and classism have also undermined vaccine uptake in people of color and lower-income people. When the vaccine was first rolled out, eligibility for the vaccine was limited to healthcare workers and persons ages 65 and older, excluding many nonelderly, low-income, essential workers, and people of color who were at high risk [14]. Lack of access to a regular health provider and accurate health information contributed to low vaccine uptake among the uninsured, even though the vaccine was free for all who could access it. Additionally, whereas initial vaccine rollout focused on centralized mass distributions sites, community-based delivery has since been shown to be more effective in reaching lower-income people and people of color [15,16]. Lack of paid time off to get vaccinated and recover, limited outreach to non-English speakers, and low trust in the health system have also been barriers [17,18]. Structural racism also drives socioeconomic differences by race/ethnicity that affect access to transportation and time to get vaccinated. Federal and state leaders began to message

that the pandemic was ending in May 2021, removing masking requirements when less than 40% of the US population was vaccinated and did not reinvigorate vaccine communications and delivery when the Delta and Omicron variants began to spread.

MA allocated funds to improve equity in COVID-19 vaccine coverage. In February 2021, the MA Vaccine Equity Initiative (VEI) aimed to improve vaccine administration rates in the 20 most disproportionately impacted communities [19]. These communities were identified based on COVID-19 case rates, CDC-defined Social Vulnerability Index, and the percentage of people of color [20]. The VEI ensured a vaccine pipeline to these communities during the initial rollout and has awarded $58 million to 198 community organizations to support vaccine education and outreach [21]. In March 2022, the state legislated a "COVID-19 vaccination equity plan. . . to eliminate disparities in the rates of vaccination" including boosters [21].

In this study, we assess variation in primary vaccine series and booster coverage across MA ZIP codes, analyzing data 18 months into MA's general population vaccine rollout. Boosters are a critical line of defense in mitigating the impacts of future waves [22,23]. Although the first year of the vaccine rollout involved extensive community outreach [16], the booster campaign (including the bivalent booster that became available in September 2022) has received less public attention despite its importance for maintaining immunity and protecting against Omicron variants. Closing vaccination gaps requires an understanding of barriers to uptake and identifying geographic areas with low coverage. ZIP codes are small enough to capture sociodemographic heterogeneity that is obscured in city, town, or county-level estimates. We additionally stratified our analysis by age, the leading risk factor for severe illness due to COVID-19 [24] and a strong correlate of vaccination rates. While racial disparities have often been attributed to vaccine hesitancy or medical mistrust, structural racism also drives large racial differences in income, education, and occupation that shape access to information, health services, and time off from work and could affect vaccination rates. We therefore assessed the extent to which socioeconomic factors mediated disparities in vaccination by ZIP code racial/ethnic composition. Our analysis complements aggregate reporting by the MA Department of Public Health (DPH) on the 20 cities and towns that have been the focus of the VEI, as well as prior reports of vaccination patterns at the city/town level and across Boston ZIP codes [25].

## Methods

### Data sources

**Vaccines and booster shots delivered.** We extracted data on counts of vaccinations by ZIP code and sex and by ZIP code and age group, published by MA DPH as "Weekly COVID-19 Municipality Vaccination Data" [26] Data are reported to the state by health facilities and vaccination sites. Residential postal ZIP codes and patient age are extracted from paperwork filed at the vaccination site. We analyzed data on vaccines administered in MA from the start of the vaccine rollout through October 10, 2022. MA rolled out the vaccine in phases, first to clinicians and nursing homes in December 2020, then to members of the general population in February 2021, starting with adults over 75 years. This study is reported as per the Strengthening the Reporting of Observational Studies in Epidemiology (STROBE) guideline (S1 STROBE Checklist). The study did not have a prespecified analysis plan.

We extracted data on two key constructs reported in the MA DPH vaccine database:

- Primary vaccine series: one shot if Ad26.COV2.S (e.g., Janssen/Johnson and Johnson), two shots if mRNA-1273 (e.g., Moderna) or BNT162b2 (e.g., Pfizer-BioNTech) vaccine; this definition corresponds with the "fully vaccinated" data reported by MA DPH.

- Booster shot (any booster shot after completing initial vaccine schedule).

As of October 2022, all MA residents ages 5 years and over were eligible to be vaccinated and to receive the booster shot if their primary series vaccine was completed at least 2 months prior.

MA DPH reports data on vaccines administered by ZIP code, stratified by age, sex, and race/ethnicity. ZIP code-level totals, which are not reported, can be constructed through aggregation. To protect confidentiality, MA DPH suppresses exact numbers in cells with fewer than 30 people vaccinated/boosted. To minimize the influence of missing data in the ZIP code-by-age data, we instead used ZIP code-by-sex data to estimate aggregate ZIP code totals for people ages 5 and older. The ZIP code-by-sex data were nearly complete for self-identified "males" and "females." Data for the category "neither male nor female" were frequently suppressed due to small numbers (75% of ZIP codes were missing data on this group for vaccines, and 96% were missing data on boosters). We imputed data for "neither male nor female" as follows. We computed the ratios of people vaccinated/boosted among persons "neither male nor female" relative to the numbers of people vaccinated/boosted among persons either "male" or "female" for those ZIP codes reporting all three categories. These ratios were 0.4% for vaccinated and 0.3% for boosted. We then used these ratios to estimate the number of people "neither male nor female" vaccinated/boosted based on the number either "male" and "female" for those ZIP codes that did not report numbers for "neither male nor female." Where the estimated number "neither male nor female" exceeded the data suppression threshold of 30, we imputed a value of 30. We then aggregated across all sexes to obtain counts of ZIP code residents who had received the primary vaccine series and/or booster shot.

In addition to ZIP code totals, we assessed ZIP code-by-age primary series vaccine and booster coverage. MA DPH reports the following age categories: 5 to 11, 12 to 15, 16 to 19, 20 to 29, 30 to 39, 40 to 49, 50 to 59, 60 to 64, 65 to 69, 70 to 74, 75+ years. Some ZIP-by-age cells were missing data due to suppression rules (0.04% of cells for vaccinated, 3.7% of cells for boosted). To impute values, we estimated a mixed effects Poisson model with fixed effects for each age group and interactions of each age group with median income, ZIP code random effects, and ACS population denominators (described below) as the offset. The model predictions were strongly correlated (0.98) with the observed values. We obtained predicted counts for the missing cells and imputed either that value or 30, whichever was smaller. We then aggregated to broader age groups for the analysis: 5 to 19, 20 to 39, 40 to 64, and 65+ years.

Approximately 3% of MA residents receiving primary series vaccines and 2% of MA residents receiving boosters did not report their ZIP code and were excluded from the analysis.

**Population denominators.** We constructed ZIP code-by-age population counts using publicly available data from the American Community Survey (ACS) 5-year combined estimates (2015 to 2019) [27]. The ACS is a random sample survey of approximately 3 million people in the US population each year conducted by the US Census Bureau. Because postal ZIP codes (reported on vaccine forms) differ from Census-defined ZIP code tabulation areas (ZCTAs), we constructed population denominators de novo, aggregating up from census tracts. Age-specific population data were extracted from ACS at the census tract level. We used the Housing and Urban Development tract to ZIP code 2019 fourth quarter crosswalk to assign these census tract populations to postal ZIP codes [28]. Most census tracts are fully contained within single ZIP codes; however, some are split across ZIP codes. For these, we allocated age-specific population counts to ZIP codes proportionately based on the populations of the underlying census blocks and their ZIP code membership. We note that sampling variability in the population denominators implies that the estimated population size may sometimes be smaller than the number of residents vaccinated.

**ZIP code characteristics.** Census tract-level ACS 5-year estimates for 2019 were obtained for race/ethnicity (B03002), median household income (B19001), educational attainment (B15003), and occupation (S2404). Essential workers were defined based on definitions developed by the American Civil Liberties Union, which identified those job types considered "essential" during the pandemic such as work in healthcare, transportation, and food preparation [29] and has been used to assess COVID-19 inequities in MA [10–12]. We aggregated to postal ZIP codes using the tract-to-ZIP crosswalk, weighting by tract population. Using the Census ACS estimates, we derived ZIP code-level "percent college graduates over the age of 25 years," "percent Black, Latino, or Indigenous," "percent essential workers," and "median household income." Race/ethnicity in the ACS was determined by self-report. People were identified as Black, Latino, or Indigenous if they reported Black or American Indian/Alaska Native among their "races" or if they reported Hispanic ethnicity. We aggregated Black, Latino, and Indigenous MA residents into a single measure due to the high concentration of each of these groups in a relatively small number of ZIP codes. Aggregation of income data yielded population-weighted averages of median household income across census tracts within each ZIP code. For convenience, we refer to this ZIP code-level measure as "median household income." We additionally constructed a series of variables to capture differences in age composition across ZIP codes: percentage of ZIP code residents aged 0 to 4 years, 5 to 19 years, 20 to 39 years, 40 to 64 years, and 65+ years. Finally, we created an indicator for whether the ZIP code was within one of the state's 20 VEI communities: Boston, Brockton, Chelsea, Everett, Fall River, Fitchburg, Framingham, Haverhill, Holyoke, Lawrence, Leominster, Lowell, Lynn, Malden, Methuen, New Bedford, Randolph, Revere, Springfield, and Worcester.

**Exclusions.** We excluded ZIP codes that did not correspond to residential areas, ZIP codes that corresponded to specific universities or businesses, and ZIP codes assigned to post office boxes and not to residential addresses. This reduced the total number of ZIP codes from 648 to 481. To avoid unstable estimates due to very small denominators, we excluded the smallest ZIP codes containing 1% of the total population ($n = 63$). After these exclusions, we had estimates of vaccination and booster coverage for all remaining ZIP codes ($n = 418$).

## Analysis

We constructed estimates of "percent vaccinated" (primary series) and "percent boosted" for MA ZIP codes and for ZIP code-by-age-group cells. Different age categories were reported in the state vaccination data and the ACS. To harmonize age groups and reduce the number of small cells, we collapsed age to the following categories: 5 to 19 years, 20 to 39 years, 40 to 64 years, and 65+ years.

Uncertainty exists when estimating vaccine coverage for specific ZIP codes due to sampling error in the population denominators. The ACS population estimates are published with standard errors; however, these are reported at the census tract (not ZIP code) level. We captured this uncertainty when reporting estimates for specific ZIP codes (and ZIP-by-age cells) by constructing 90% confidence intervals (CIs) using a resampling approach. For each census tract by age group cell, we simulated 101 population estimates from a normal distribution defined by the ACS point estimates and standard error for that observation. We then aggregated to ZIP codes and computed the constructs of interest holding the numerator constant. To capture uncertainty in the estimates, we then ranked these simulated estimates and used the fifth and 95th ranked estimate as the lower- and upper-bound for our 90% CI.

**ZIP code-level analysis.** We assessed spatial heterogeneity in vaccine coverage by mapping the percent of residents ages 5 years and older who have received the primary series vaccine or booster shot. We created higher-resolution maps for densely populated areas. We then

investigated the association of ZIP code vaccination and booster coverage with ZIP code characteristics: median household income, percent college graduates, percent Black/Latino/Indigenous, percent essential worker, and percent in each age group: 5 to 19, 20 to 39, 40 to 64, and 65+ years. We present scatter plots and estimated bivariate linear regression models with robust standard errors.

We also estimated multivariable regression models to determine whether the bivariate associations are statistically explained by other factors. For example, higher-income MA residents are on average older, which could lead to a positive association between ZIP code income and vaccination rates. Of particular interest, we assessed whether percent Black, Latino, and Indigenous was independently associated with vaccine coverage, after adjusting for age shares and after adjusting for income, education, and essential worker shares—factors that may influence access to vaccination. If vaccine hesitancy related to racial/ethnic identity constrained uptake in communities of color, then we would expect to see a persistent negative association between percent Black, Latino, and Indigenous and vaccine uptake in these adjusted models.

**ZIP code-by-age analysis.** To illustrate heterogeneity in age-specific primary series vaccination and booster rates, we plotted ranked coverage rates, ordered by ZIP code. In these distribution plots, we censored the top and bottom 5% of observations to facilitate visualization of the rest of the distribution. To assess the association between vaccination rates and ZIP code population characteristics, we constructed population-weighted deciles for each characteristic. We then computed binned averages and 95% CIs within each decile to estimate the percent vaccinated and percent boosted. We also assessed the association of these ZIP code characteristics with percent vaccinated and boosted in age-stratified multivariable regression models, allowing for different relationships within each age group. To test for effect heterogeneity, we fit regression models on all age groups, interacted the covariate of interest with age, and tested the null hypothesis that the coefficients on the interaction terms were jointly equal to zero.

**Analysis plan.** As the study did not have a prespecified analysis plan, we describe the timing of our analytic choices here. The socioeconomic and demographic variables used in the assessment of vaccine uptake disparities were selected prior to the analysis and were chosen based on the prior literature on disparities in COVID-19 impact and early vaccine uptake. The approach to constructing population denominators followed prior work by the research team. The age ranges used in the analysis were determined a priori based on harmonization of ages reported in the state vaccination data and ACS denominator data. The decision to exclude ZIP codes with small populations was made after observing the high rates of data suppression and instability (wide CIs) of estimates in those ZIP codes. The methods for the decile-based descriptive analysis and multiply-adjusted regression analyses were planned a priori. At the suggestion of a reviewer, we tested the null hypothesis that the VEI had the same association with vaccine coverage as with booster coverage, given that community outreach was strongest during the initial vaccine rollout. We also assessed associations stratified by male/female sex.

**Ethics.** The Boston University Medical Campus IRB does not require ethical review for secondary analyses of publicly available, deidentified data.

## Results

We analyzed data on 418 ZIP codes containing 97% of the MA population (Table 1). Of these ZIP codes, 184 (44%) were in cities or towns containing multiple ZIP codes. The mean population of the included ZIP codes was 15,967 (range 2,014 to 61,099) individuals. On average, the study population resided in ZIP codes where 41% of residents were college educated, 19% were Black, Latino, or Indigenous, 32% were essential workers, and where median household income was $41,100.

**Table 1. Characteristics of the study population.**

| | |
|---|---|
| ZIP codes (N) | 418 |
| Total population (N) | 6,674,243 |
| ZIP code population (mean) | 15,967 |
| ZIP code population (range) | 2,014 to 61,099 |
| *Percentage with a primary series vaccine* | *Mean (SD)* |
| Aged 5+ years | 83% (10%) |
| 5–19 years | 65% (17%) |
| 20–39 years | 79% (14%) |
| 40–64 years | 85% (10%) |
| 65+ years | 100% (10%) |
| *Percentage with a booster shot* | *Mean (SD)* |
| Aged 5+ years | 50% (11%) |
| 5–19 years | 24% (12%) |
| 20–39 years | 39% (12%) |
| 40–64 years | 55% (11%) |
| 65+ years | 81% (10%) |
| *ZIP code characteristics* | *Mean (SD)* |
| Median household income | $41,100 ($12,500) |
| % college graduates | 41% (17%) |
| % Black, Latino, or Indigenous | 19% (20%) |
| % essential workers | 32% (6%) |

**Note**: Mean and standard deviation (SD) of vaccination coverage and ZIP code characteristics are weighted by ZIP code population ages 5 years and older. Percent with a primary series vaccine and booster shot are as of October 10, 2022.

On average, an estimated 83% (5,239,410/6,321,016) of residents ages 5 years and older had received the primary series vaccine, and 50% (3,163,944/6,321,016) had received a booster shot. We estimated that 100% (1,079,095/1,074,418) of residents ages 65 years and older had received the primary series vaccine. (We have not constrained the numerator to be less than the denominator in these estimates.) Coverage for the primary series vaccine was lower in younger age groups, with 85% (1,888,624/2,223,264) of persons 40 to 64 years, 79% (1,444,242/1,837,095) of persons 20 to 39 years, and 65% (765,259/1,186,227) of persons 5 to 19 years vaccinated. Booster coverage was 81% (871,749/1,074,418) among residents 65 years and older, 55% (1,230,146/2,223,264) among persons 40 to 64 years, 39% (725,306/1,837,095) among persons 20 to 39 years, and 24% (286,712/1,186,227) among persons 5 to 19 years.

The maps in Fig 1 show ZIP code-level variation in vaccination (top) and booster (bottom) coverage for residents ages 5 years and older. The panels show similar geographic patterns of vaccine and booster uptake, with the highest coverage among ZIP codes in Boston's Western suburbs, the North and South Shore, Cape Cod and the Islands, and in the college towns of the Pioneer Valley. Lower vaccine and booster coverage was observed in lower-income ZIP codes of the urban centers—greater Boston, Worcester, Springfield, New Bedford, Fall River, Lawrence, and Lowell (S1 Fig shows higher-resolution maps of these areas). Additionally, lower coverage was observed in nonurban ZIP codes in Central, Western, and Southeast MA, and in MA ZIP codes along the borders with New Hampshire, Rhode Island, and Connecticut.

Fig 2 shows scatter plots of vaccine and booster coverage against ZIP code characteristics. We observed strong correlations between primary series vaccine and booster coverage and the four sociodemographic indicators. A $10,000 increase in a ZIP code's median household

## (A) Percent vaccinated

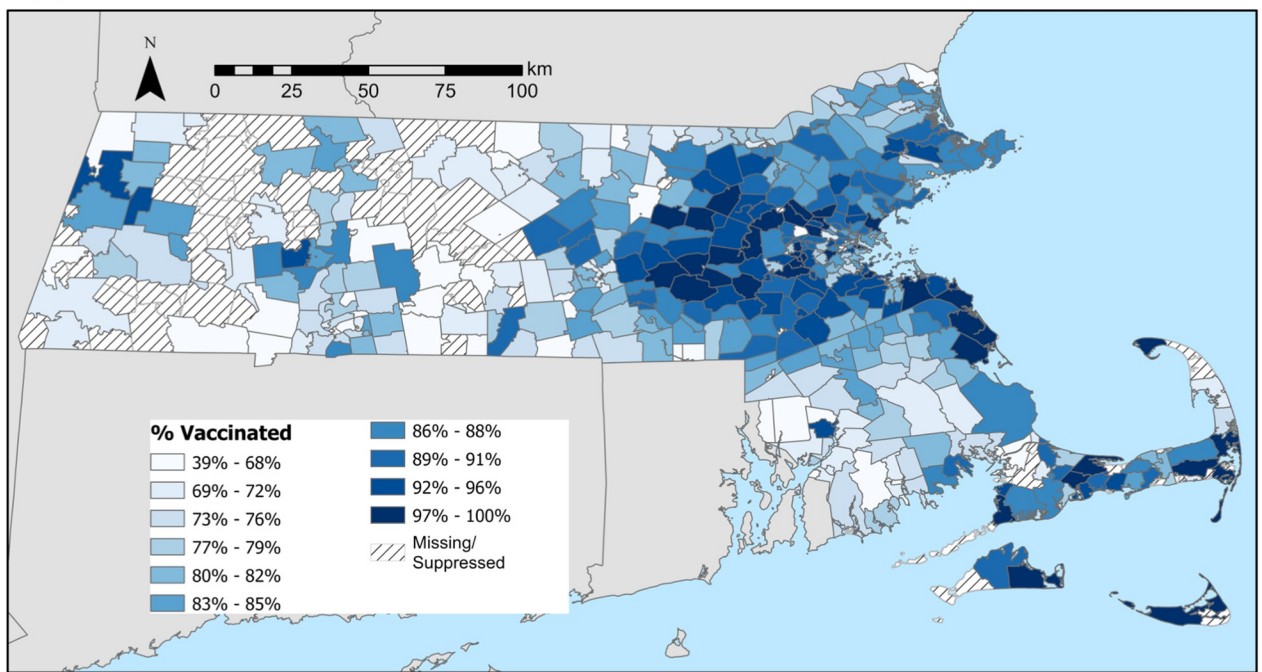

## (B) Percent boosted

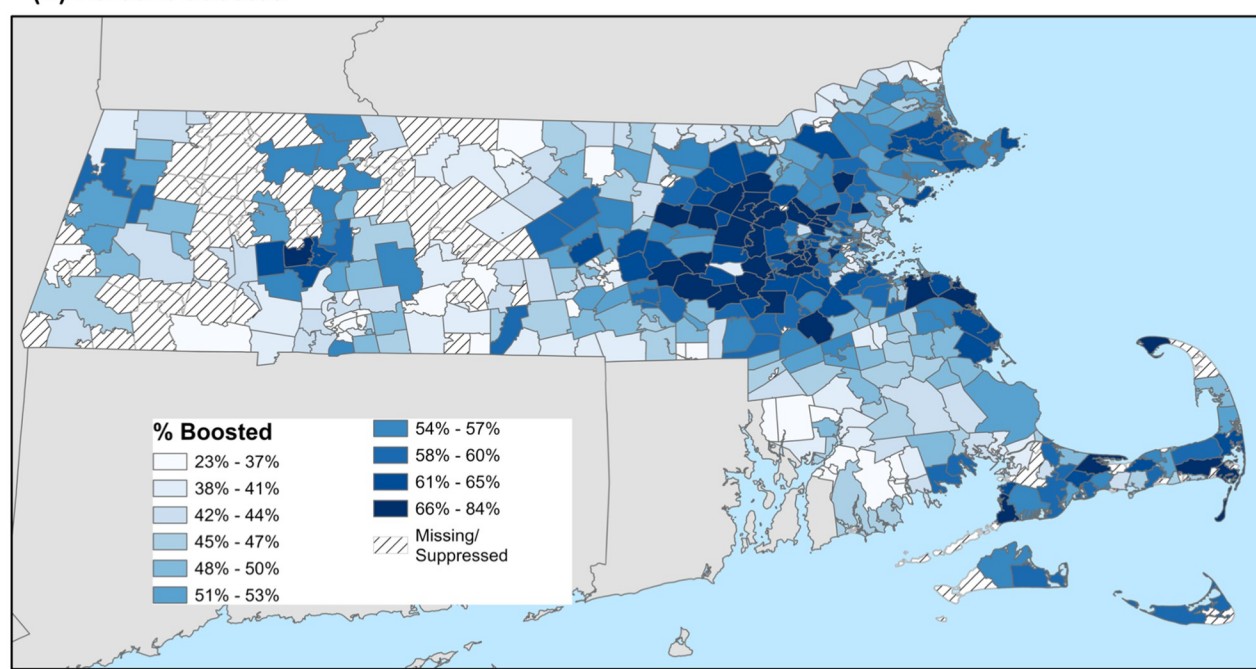

**Fig 1. Percentage of residents with a COVID-19 (a) vaccine or (b) booster by ZIP code as of October 10, 2022. Note**: We analyzed data on 418 ZIP codes containing 97% of the MA population. ZIP codes labeled "missing/suppressed" were excluded because they corresponded to specific businesses or universities, to post office boxes rather than residential addresses, or because were the smallest 1% of ZIP codes, which we excluded due to instability of estimates. To facilitate comparisons across ZIP codes within each panel (**a**) and (**b**), the scales in the panels differ. Shape files were obtained from the US Census Bureau, accessed October 20, 2022: https://www.census.gov/geographies/mapping-files/time-series/geo/tiger-geodatabase-file.html. Copyright protection is not available for any work of the United States Government (Title 17 U.S.C., Section 105).

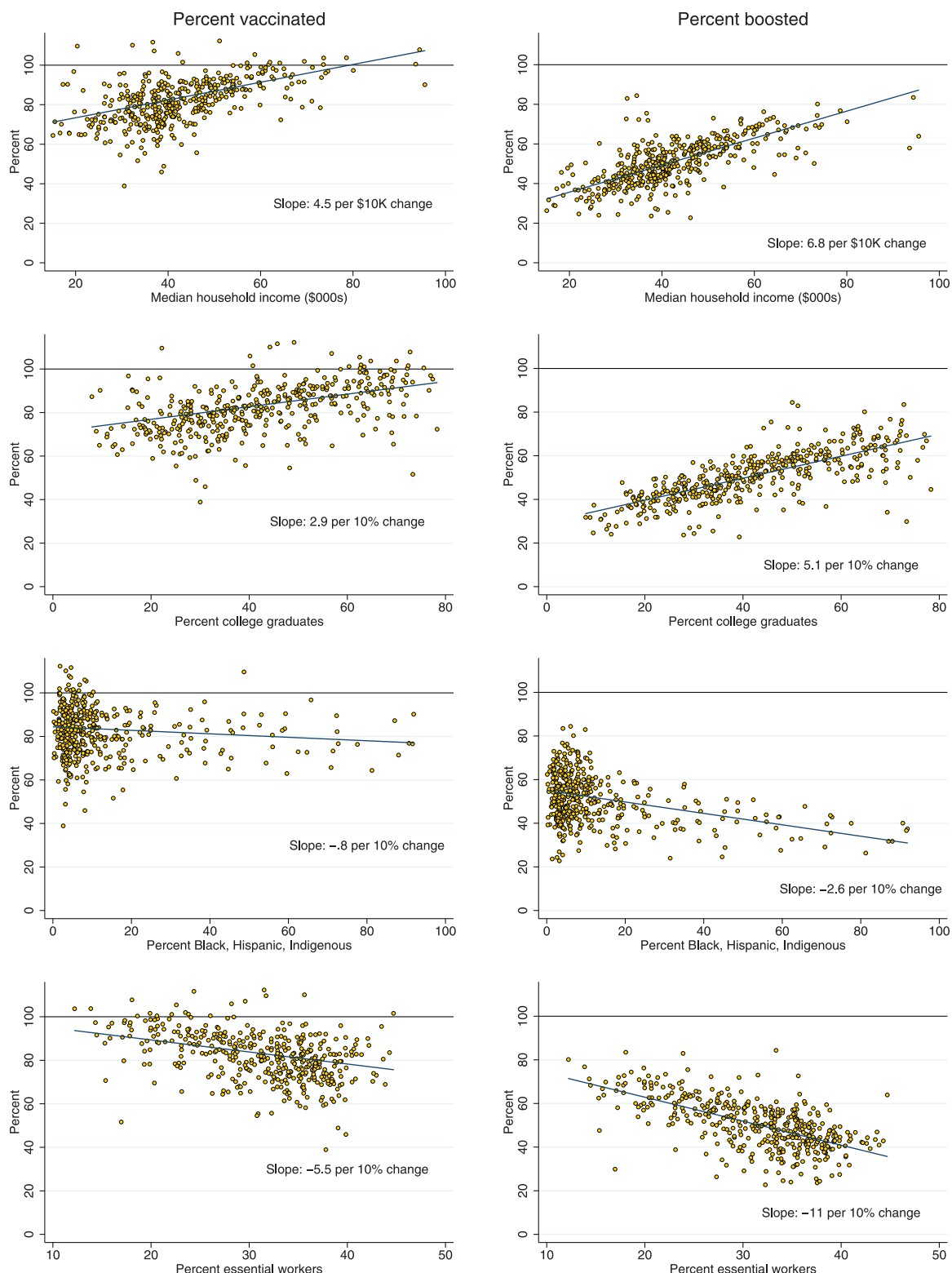

**Fig 2. Percent vaccinated and boosted according to individual ZIP code characteristics. Note**: Data on vaccine and booster coverage are as of October 10, 2022. Fitted lines are from bivariate regressions. Coefficients and confidence intervals for bivariate and multivariable models are presented in Table 2. To preserve the scale across the plots, three outliers with estimated vaccine coverage over 115% were suppressed in the scatter plots but contribute to the fitted lines.

income was associated with a 4.5 percentage point (95% CI 3.8 to 5.2, $p < 0.001$) increase in primary series vaccine coverage and a 6.8 percentage point (95% CI 6.2 to 7.4, $p < 0.001$) increase in booster coverage. From the lowest to the highest income levels, booster coverage increased from about 30% to over 70%. A 10 percentage point increase in percent college graduates was associated with a 2.9 percentage point (95% CI 2.2 to 3.6, $p < 0.001$) increase in vaccinations and a 5.1 percentage point (95% CI 4.5 to 5.6, $p < 0.001$) increase in boosters. A 10 percentage point increase in percent essential workers was associated with a −5.5 percentage point (95% CI −7.3 to −3.8, $p < 0.001$) reduction in primary series vaccination coverage and a −11.0 percentage point (95% CI −12.4 to −9.6, $p < 0.001$) reduction in booster coverage. We observed a weaker relationship between percent Black, Latino, and Indigenous and vaccination coverage, although a 10 percentage point increase in this population share was associated with a −2.6 percentage point (95% CI −3.0 to −2.2, $p < 0.001$) decline in booster coverage.

Table 2 shows results from both bivariate and multivariable models regressing percent vaccinated/boosted on variables describing the age composition of the ZIP code (defined above) as well as the four sociodemographic characteristics. In these multivariable regression models, percent college graduates emerged as the strongest predictor of vaccine and booster coverage. Each 10 percentage point increase in percent college graduates was associated with a 5.1 percentage point (95% CI 3.9 to 6.3, $p < 0.001$) increase in vaccine coverage and a 5.4 percentage point (95% CI 4.5 to 6.4, $p < 0.001$) increase in booster coverage. Although median household income was strongly positively associated with vaccination and booster rates in the multivariable analyses, these associations were substantially attenuated and lost statistical significance after adjusting for age composition, suggesting that age distribution confounds the relationship between ZIP code median income and vaccination and booster rates when controlling for other factors (S1 Table).

We observed notable changes in the coefficients for "percent Black, Latino, or Indigenous" and "percent essential workers" after adjusting for age composition and education. In bivariate models, these ZIP code characteristics were negatively associated with percent vaccinated; however, in multivariable models, the associations turned positive. ZIP codes with a larger share of Black, Latino, and Indigenous residents and with more essential workers had higher vaccination rates than would be otherwise expected based on the education and age composition in those ZIP codes. Adjusting for age and education, each 10 percentage point increase in "percent Black, Latino, or Indigenous" was associated with a 1.9 percentage point (95% CI 1.0 to 2.8, $p < 0.001$) increase in vaccine coverage and each 10 percentage point increase in "percent essential workers" was associated with a 4.8 percentage point (95% CI 2.6 to 7.1, $p < 0.001$) increase in vaccine coverage. For boosters, while the associations from the bivariate models were similarly attenuated after adjusting for age and education, there was a less pronounced association in the multivariable models with percent essential workers (null) and percent Black, Latino, or Indigenous (modestly positive).

Table 2 includes an indicator for whether the ZIP code was a part of one of the 20 VEI communities targeted by the state for enhanced vaccine outreach. VEI communities had lower vaccine coverage (−4.0 percentage points, 95% CI −6.4 to −1.6, $p < 0.01$) and booster coverage (−11.1 percentage points, 95% CI −13.4 to −8.8, $p < 0.001$) than non-VEI communities. After adjusting for sociodemographic characteristics and age composition, the coefficient on VEI status was close to zero: −0.3 percentage points (95% CI −3.6 to 3.0, $p = 0.85$) for vaccine coverage and −2.1 percentage points (95% CI −4.4 to 0.2, $p = 0.079$) for booster coverage. The VEI coefficients in the adjusted vaccine and booster models were significantly different ($p = 0.049$). (We note that other factors not included in our model may explain the lower vaccination or booster rates in VEI communities and that vaccination and booster rates might have been

**Table 2. Association of ZIP code characteristics with the percent of residents aged 5 years and older that are vaccinated for COVID-19 and/or have had a COVID-19 booster shot.**

**a) Percent vaccinated**

| OLS regression models | Bivariate | | Multivariable | | | | | |
| --- | --- | --- | --- | --- | --- | --- | --- | --- |
| | (1) | | (2) | | (3) | | (4) | |
| | beta (95% CI) | p-value | beta (95% CI) | p-value | beta (95% CI) | p-value | beta (95% CI) | p-value |
| Median household income ($10k increase) | 4.5*** (3.8,5.2) | <0.001 | 4.8*** (3.3,6.3) | <0.001 | 1.5 (−0.1,3.1) | 0.060 | 1.5 (−0.1,3.1) | 0.059 |
| Percent college graduates (10% point increase) | 2.9*** (2.2,3.6) | <0.001 | 2.3*** (1.1,3.6) | <0.001 | 5.1*** (3.9,6.3) | <0.001 | 5.1*** (3.9,6.3) | <0.001 |
| Percent Black, Latino, Indigenous (10% point increase) | −0.8** (−1.3,−0.3) | 0.002 | 1.4*** (0.8,2.0) | <0.001 | 1.8*** (1.1,2.6) | <0.001 | 1.9*** (1.0,2.8) | <0.001 |
| Percent essential worker (10% point increase) | −5.5*** (−7.3,−3.8) | <0.001 | 4.5*** (2.1,6.8) | <0.001 | 4.8*** (2.6,7.1) | <0.001 | 4.8*** (2.6,7.1) | <0.001 |
| VEI community (0,1) | −4.0** (−6.4,−1.6) | 0.001 | | | | | −0.3 (−3.6,3.0) | 0.850 |
| Adjusted for age distribution | No | | No | | Yes | | Yes | |
| $R^2$ | NA | | 0.38 | | 0.47 | | 0.47 | |

**b) Percent with a booster shot (dependent variable)**

| OLS regression models | Bivariate | | Multivariable | | | | | |
| --- | --- | --- | --- | --- | --- | --- | --- | --- |
| | (1) | | (2) | | (3) | | (4) | |
| | beta (95% CI) | p-value | beta (95% CI) | p-value | beta (95% CI) | p-value | beta (95% CI) | p-value |
| Median household income ($10k increase) | 6.8*** (6.2,7.4) | <0.001 | 3.7*** (2.2,5.2) | <0.001 | 0.6 (−0.8,2.0) | 0.392 | 0.6 (−0.7,1.9) | 0.351 |
| Percent college graduates (10% point increase) | 5.1*** (4.5,5.6) | <0.001 | 2.8*** (1.6,4.1) | <0.001 | 5.5*** (4.5,6.4) | <0.001 | 5.4*** (4.5,6.4) | <0.001 |
| Percent Black, Latino, Indigenous (10% point increase) | −2.6*** (−3.0,−2.2) | <0.001 | −0.1 (−0.4,0.3) | 0.687 | 0.8*** (0.4,1.2) | <0.001 | 1.0*** (0.5,1.5) | <0.001 |
| Percent essential worker (10% point increase) | −11.0*** (−12.4,−9.6) | <0.001 | 0.2 (−1.7,2.2) | 0.801 | 0.3 (−1.3,1.9) | 0.714 | 0.4 (−1.2,1.9) | 0.630 |
| VEI community (0,1) | −11.1*** (−13.4,−8.8) | <0.001 | | | | | −2.1 (−4.4,0.2) | 0.079 |
| Adjusted for age distribution | No | | No | | Yes | | Yes | |
| $R^2$ | NA | | 0.67 | | 0.76 | | 0.76 | |

**Note**: $N$ = 418 ZIP codes. In column (1), each <u>cell</u> shows the result of a separate bivariate OLS regression model. In columns (2)–(4), each <u>column</u> shows the results of multivariable (adjusted) OLS regression models. Coefficients are scaled so that they can be interpreted as the percentage point change in primary series (or booster) coverage associated with a 10% (or $10K) increase in the continuous covariates. 95% CIs based on heteroskedasticity-robust standard errors are shown in parentheses. We adjusted for age by including variables denoting the percentage of ZIP code residents in the following age groups: 5–19, 20–39, 40–64, and 65+ years.

*$p < 0.05$,

**$p < 0.01$,

***$p < 0.001$.

CI, confidence interval; OLS, ordinary least squares; VEI, Vaccine Equity Initiative.

even lower in the absence of the VEI.) In sex-stratified models, results were similar for men and women (S2 Table).

We also evaluated whether ZIP code inequities in vaccination coverage varied by age group. Fig 3 (left-sided graphs) shows the distribution of "percent vaccinated" in each ZIP code, stratified by age group. The point estimates contain estimation uncertainty in the denominator, which explains why some percentages exceed 100%. The figure shows very high vaccination rates among the elderly (65+ years), with relatively little variation across ZIP codes. Variation in primary series vaccine coverage was more pronounced for younger age groups, in particular, for children (5 to 19 years), where vaccination rates vary from as low as 40% to as high as 90%. Fig 3 (right-sided graphs) shows "percent boosted" in each ZIP code, stratified by age group. The share boosted varied widely in all age groups, even for the elderly. For people over 65 years, the share boosted varied from approximately 60% in the lowest coverage ZIP codes to nearly 100% in the highest-coverage ZIP codes.

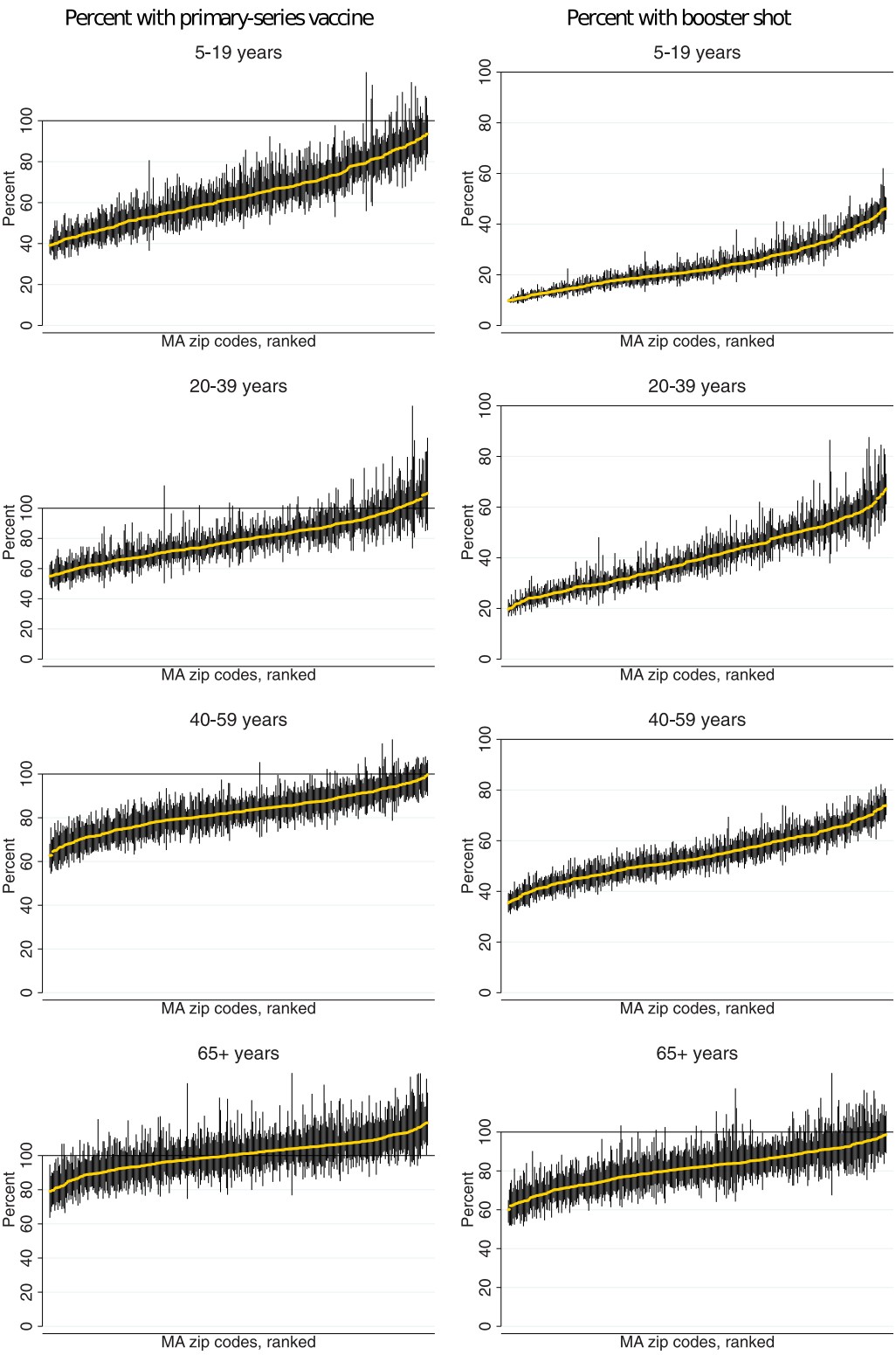

**Fig 3. Percentage of residents who have received a COVID-19 vaccine by ZIP code and age. Note**: Data are ranked by percent vaccinated with the primary series or booster shot. Vertical lines are 90% confidence intervals, which capture uncertainty due to sampling error in the population denominators. Due to the presence of sampling error in the denominators, neither the point estimates nor confidence bounds are constrained to be within the [0%,100%] interval.

Fig 4 shows the share of MA adults by age group who are vaccinated and boosted by median household income and by the percent of ZIP code residents that are college graduates, Black, Latino, or Indigenous, and essential workers. MA has achieved very high vaccination coverage among 65+-year-olds across ZIP codes with different population characteristics (bottom graphs of the figure). Vaccination coverage among MA residents 65+ years was above 95% in all deciles of all four ZIP code characteristics, and close to 100% for most. Vaccination coverage among adults ages 40 to 64 years was also relatively high and equitably distributed (third set of graphs of the figure). At least 75% of residents were vaccinated in all ZIP code covariate deciles, and vaccination rates were *highest* in ZIP codes with a greater share of Black, Latino, or Indigenous residents. Larger gaps (and inequities) in vaccine coverage were apparent for MA adults ages 20 to 39 (second set of graphs) and children ages 5 to 19 (top graphs of the figure).

By comparison, there was wide variation in coverage of the booster shot across all age groups. The share of MA residents over 65 years who had received a booster was under 80% in ZIP codes with the lowest income, lowest share of college graduates, and highest shares of Black, Latino, or Indigenous residents and essential workers. Patterns were similar for middle-aged adults (40 to 64 years), with pronounced disparities and rates under 50% for ZIP codes in the lowest deciles of education and income and the highest deciles of Black, Latino, or Indigenous and essential worker shares. Booster coverage was nearly 70% in higher-income ZIP codes. Large disparities in booster coverage were also apparent for younger adults and children, with gaps in coverage of over 30 percentage points between the top and bottom deciles of ZIP code income and gaps over 20 percentage points by deciles of education and essential worker share.

Table 3 presents age-stratified bivariate and multivariable regression models, replicating Table 2 for each age group—5 to 19, 20 to 39, 40 to 64, and 65+ years. Inequities in vaccine and booster uptake were most pronounced in younger age groups. In the bivariate models (panels (a) and (c)), income and education had positive associations with vaccine and booster coverage across all age groups, although the gradient was steeper at younger ages. Percent Black, Latino, or Indigenous was negatively associated with vaccine coverage among children and younger adults and with booster coverage at all ages, but positively associated with vaccine coverage among adults age 40 to 64. Percent essential workers was negatively associated with vaccine and booster coverage for all age groups. For all ZIP code characteristics, we rejected the null hypothesis of effect homogeneity across age groups (F-test, $p < 0.0001$).

In multivariable models (panels (b) and (d)), ZIP code income was strongly associated with vaccine and booster coverage for children (5 to 19 years) and young adults (20 to 39 years), but not for older ages. Education levels maintained strong positive associations across all age groups. After adjusting for income and education, ZIP codes with larger Black, Latino, or Indigenous populations had similar or higher vaccination coverage across all age groups, in contrast to the bivariate models. In the adjusted models, residing in a VEI community was not associated with vaccine or booster coverage in any age group except for persons 65+ years who were −3.7 percentage points (95% CI −7.3 to −0.1, $p < 0.05$) less likely to be boosted if they lived in a VEI community.

## Discussion

We assessed geographic and sociodemographic equity in the state of Massachusetts (MA)'s COVID-19 vaccination program as of October 10, 2022, through analysis of newly released data on primary series vaccinations and boosters by age and ZIP code. Our findings indicate that coverage of the primary series vaccine among elderly MA residents (65 years and older)

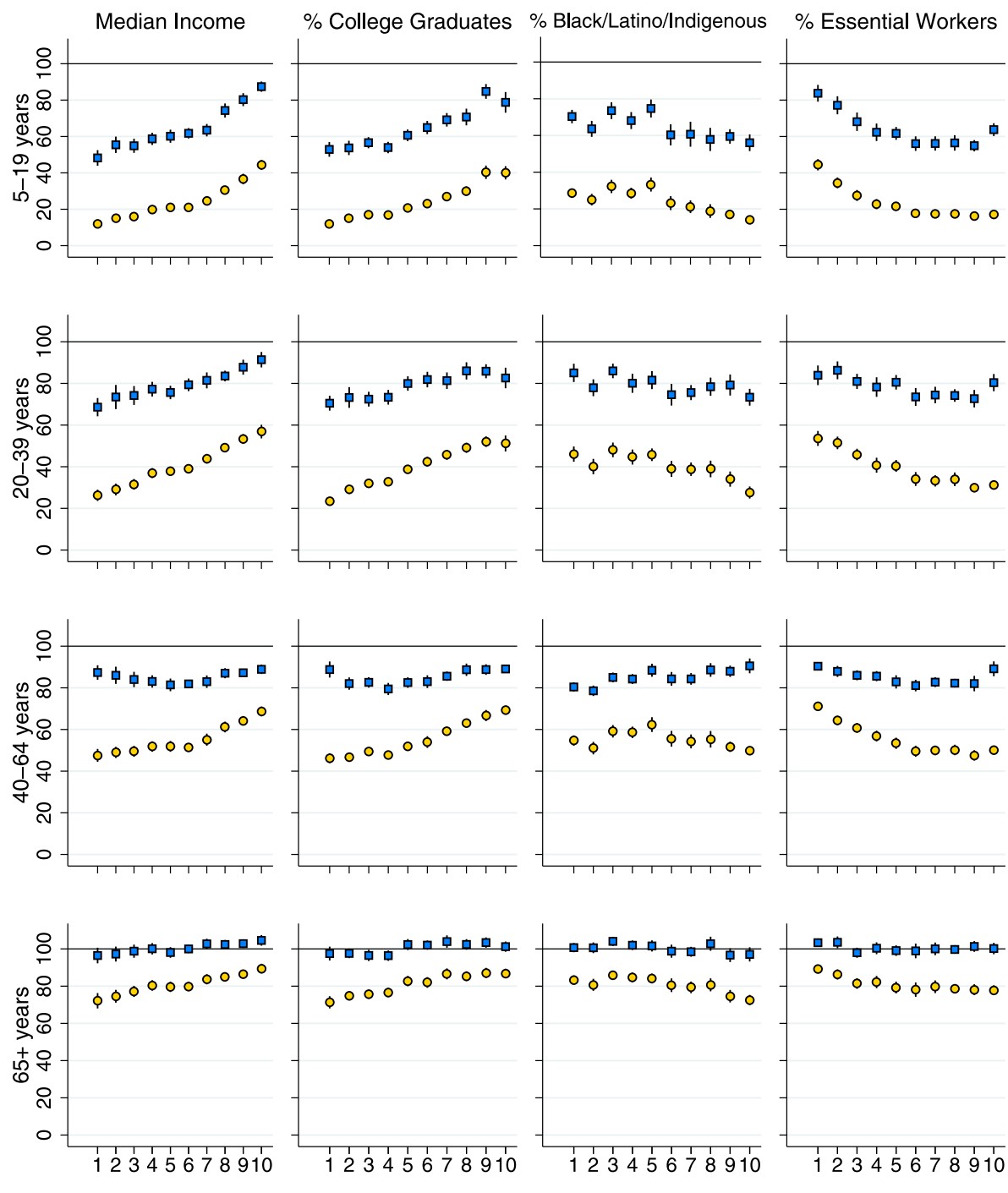

**Fig 4. Percent vaccinated or boosted in each age group, by decile of ZIP-code characteristics. Note**: Plot displays % vaccinated (blue squares) and % boosted (yellow circles) for MA residents ages 5–19, 20–39, 40–64, and 65+ years, displayed by decile of four ZIP code characteristics. The first decile is the bottom 10% of ZIP codes, ranked by each covariate; the 10th decile is the top 10% of ZIP codes. Vertical spikes are 95% confidence intervals. Due to sampling error in the population denominators, neither point estimates nor CIs are constrained to be below 100%.

**Table 3. Association of ZIP code characteristics with percent vaccinated and boosted, by age.**

**a) Percent vaccinated: Age-stratified bivariate OLS regression models**

| Age group | 5–19 years | | 20–39 years | | 40–64 years | | 65+ years | |
|---|---|---|---|---|---|---|---|---|
| Parameters | beta (95% CI) | p-value | beta (95% CI) | p-value | beta (95% CI) | p-value | beta (95% CI) | p-value |
| Median household income ($10k increase) | 9.2*** (8.1,10.2) | <0.001 | 5.3*** (4.1,6.5) | <0.001 | 0.8* (0.0,1.6) | 0.040 | 1.7*** (0.9,2.6) | <0.001 |
| % college graduates (10% point increase) | 5.8*** (4.7,6.9) | <0.001 | 2.9*** (1.9,3.9) | <0.001 | 1.1** (0.5,1.8) | 0.001 | 1.3*** (0.7,1.9) | <0.001 |
| % Black, Latino, Indigenous (10% point increase) | −2.1*** (−2.9,−1.3) | <0.001 | −1.0** (−1.7,−0.3) | 0.006 | 1.4*** (0.8,1.9) | <0.001 | −0.8* (−1.4,−0.2) | 0.010 |
| % essential worker (10% point increase) | −12.6*** (−15.5,−9.7) | <0.001 | −4.6** (−7.6,−1.6) | 0.003 | −2.8*** (−4.3,−1.2) | 0.001 | −1.5* (−2.9,−0.1) | 0.042 |
| VEI community (0,1) | −11.5*** (−15.5,−7.5) | <0.001 | −3.0 (−6.3,0.3) | 0.077 | 3.9*** (1.6,6.3) | 0.001 | −3.4** (−5.9,−0.9) | 0.008 |

**b) Percent vaccinated: Age-stratified multivariable OLS regression models**

| Age group | 5–19 years | | 20–39 years | | 40–64 years | | 65+ years | |
|---|---|---|---|---|---|---|---|---|
| Parameters | beta (95% CI) | p-value | beta (95% CI) | p-value | beta (95% CI) | p-value | beta (95% CI) | p-value |
| Median household income ($10k increase) | 10.1*** (7.9,12.3) | <0.001 | 6.1*** (3.9,8.3) | <0.001 | −0.2 (−1.7,1.3) | 0.759 | 1.2 (−0.7,3.0) | 0.221 |
| % college graduates (10% point increase) | 2.5** (0.7,4.3) | 0.008 | 2.7*** (1.2,4.2) | <0.001 | 3.6*** (2.2,5.1) | <0.001 | 2.1* (0.4,3.8) | 0.015 |
| % Black, Latino, Indigenous (10% point increase) | 2.8*** (1.8,3.8) | <0.001 | 1.5** (0.5,2.5) | 0.004 | 2.8*** (1.8,3.8) | <0.001 | 0.4 (−0.7,1.4) | 0.481 |
| % essential worker (10% point increase) | 4.1** (1.0,7.3) | 0.010 | 7.1*** (3.4,10.8) | <0.001 | 2.0 (−0.8,4.7) | 0.162 | 4.7** (1.5,7.9) | 0.004 |
| VEI community (0,1) | −2.9 (−7.7,1.9) | 0.232 | 0.1 (−4.7,5.0) | 0.962 | −0.7 (−4.4,3.0) | 0.707 | −1.9 (−6.0,2.2) | 0.353 |
| $R^2$ | 0.57 | | 0.27 | | 0.25 | | 0.07 | |

**c) Percent boosted: Age-stratified bivariate OLS regression models**

| Age group | 5–19 years | | 20–39 years | | 40–64 years | | 65+ years | |
|---|---|---|---|---|---|---|---|---|
| Parameters | beta (95% CI) | p-value | beta (95% CI) | p-value | beta (95% CI) | p-value | beta (95% CI) | p-value |
| Median household income ($10k increase) | 7.6*** (7.0,8.2) | <0.001 | 7.7*** (6.8,8.7) | <0.001 | 5.3*** (4.6,5.9) | <0.001 | 3.8*** (2.9,4.6) | <0.001 |
| % college graduates (10% point increase) | 5.5*** (4.9,6.1) | <0.001 | 5.5*** (4.8,6.2) | <0.001 | 4.8*** (4.3,5.2) | <0.001 | 3.1*** (2.6,3.6) | <0.001 |
| % Black, Latino, Indigenous (10% point increase) | −2.4*** (−2.9,−2.0) | <0.001 | −2.6*** (−3.1,−2.1) | <0.001 | −1.0*** (−1.4,−0.7) | <0.001 | −1.8*** (−2.3,−1.3) | <0.001 |
| % essential worker (10% point increase) | −13.3*** (−15.0,−11.6) | <0.001 | −11.9*** (−14.4,−9.5) | <0.001 | −11.1*** (−12.3,−9.9) | <0.001 | −5.5*** (−6.7,−4.2) | <0.001 |
| VEI community (0,1) | −11.9*** (−14.3,−9.6) | <0.001 | −9.2*** (−12.2,−6.2) | <0.001 | −5.4*** (−7.7,−3.1) | <0.001 | −7.7*** (−10.1,−5.4) | <0.001 |

**d) Percent boosted: Age-stratified multivariable OLS regression models**

| Age group | 5–19 years | | 20–39 years | | 40–64 years | | 65+ years | |
|---|---|---|---|---|---|---|---|---|
| Parameters | beta (95% CI) | p-value | beta (95% CI) | p-value | beta (95% CI) | p-value | beta (95% CI) | p-value |
| Median household income ($10k increase) | 5.5*** (4.2,6.7) | <0.001 | 4.4*** (3.1,5.8) | <0.001 | −0.2 (−1.5,1.1) | 0.783 | 0.7 (−1.1,2.5) | 0.433 |
| % college graduates (10% point increase) | 1.5** (0.5,2.5) | 0.005 | 3.9*** (3.1,4.7) | <0.001 | 5.7*** (4.6,6.7) | <0.001 | 3.8*** (2.2,5.4) | <0.001 |
| % Black, Latino, Indigenous (10% point increase) | 1.1*** (0.6,1.5) | <0.001 | 0.8* (0.2,1.4) | 0.011 | 1.9*** (1.3,2.5) | <0.001 | 0.3 (−0.4,1.1) | 0.407 |
| % essential worker (10% point increase) | −3.1*** (−4.9,−1.3) | <0.001 | 1.6 (−0.8,4.1) | 0.194 | −0.6 (−2.4,1.2) | 0.519 | 4.1** (1.4,6.9) | 0.004 |
| VEI community (0,1) | −2.3 (−4.9,0.2) | 0.074 | −2.1 (−5.5,1.2) | 0.213 | −2.3 (−5.1,0.6) | 0.119 | −3.7* (−7.3,−0.1) | 0.044 |
| $R^2$ | 0.76 | | 0.67 | | 0.62 | | 0.27 | |

**Note**: N = 418. In panels (a) and (c), each <u>cell</u> shows the coefficient estimate from a separate bivariate OLS regression model. In panels (b) and (d), each <u>column</u> shows adjusted coefficients from multivariable regression models. Heteroskedasticity-robust standard errors are shown in parentheses.

*p < 0.05,

**p < 0.01,

***p < 0.001.

OLS, ordinary least squares; VEI, Vaccine Equity Initiative.

was high and equitable, reaching over 95% in the lowest-income decile of ZIP codes and in those with the greatest share Black, Latino, and Indigenous. This finding likely reflects the state's consistent emphasis on vaccinating older adults, who are at high risk for severe complications. Primary series vaccine coverage for younger MA residents was lower and exhibited large disparities by ZIP code-level education, income, percent essential workers, and racial composition. In addition, very large inequities were observed for booster coverage, with gaps

of more than 30 percentage points between the lowest- and highest-income ZIP codes. These observed gaps in vaccine-induced protection could have significant adverse health consequences during future waves of COVID-19 infection. Protection against the Omicron variant is diminished in the absence of a booster [30,31].

Racial disparities in vaccination coverage were statistically explained by differences in age and education levels. Despite lower vaccination coverage in communities with large shares of essential workers and Black, Latino, or Indigenous residents, we found that these characteristics were not independently associated with lower vaccine or booster uptake. Far from "vaccine hesitancy" [32], these findings suggest greater demand for vaccination among populations that have been most affected by COVID, after adjusting for age and socioeconomic factors. A causal interpretation of our model would imply that ZIP codes with many essential workers or Black/Latino/Indigenous residents had lower vaccination or booster rates because their populations were on average younger, had lower levels of educational attainment and lower incomes, and because the vaccination campaign did not adequately address these barriers to access.

Medical mistrust was widely noted early in the rollout, and efforts at community outreach may have been important in increasing vaccine uptake among Black, Latino, and Indigenous populations. The high density of community organizations in MA may have facilitated diffusion of this information. Additionally, MA's communities of color were hit hardest by the pandemic in 2020, and fear of COVID-19 was higher in these communities, e.g., resulting in higher support for remote schooling among Black MA residents. These factors might explain why demand for COVID-19 vaccination has been high in MA's communities of color—after adjusting for age, income, and education levels. The more substantial disparities in boosters may relate to reduced outreach efforts subsequent to the initial vaccine rollout. In addition, demand for boosters could have been lower among communities and subpopulations with higher rates of infection, given the perception that acquired immunity (with or without vaccination) is sufficiently protective.

Epidemiological analyses of race/ethnicity demand attention to underlying causal pathways [33]. Differences in education, income, and occupation between racial groups are a product of structural racism. (In MA, for example, reliance on local property taxes for education funding perpetuates schooling inequities and residential segregation.) Therefore, it is appropriate to think of these variables as mediators rather than as confounders of the relationship between racial composition and vaccine coverage. Disparities by racial composition should not be ignored even if they can be statistically explained by socioeconomic factors.

Additionally, Black, Latino, and Indigenous residents and essential workers have experienced disproportionate infection rates, morbidity, and mortality from COVID-19 [10–12] and would therefore be expected to benefit more from vaccination than other groups that are less likely to be exposed to COVID and/or less likely to experience severe illness.

Along with racial/ethnic patterns, substantial disparities related to income and education attainment persist, and local health departments continue to call for more long-term investments to close these gaps. Interventions to address vaccine access barriers related to education and poverty—including offering more convenient clinic times, paid sick leave for potential side effects, providing information through trusted community-based sources in multiple languages, and conducting outreach outside of the formal medical sector—will be essential to improving equity across multiple dimensions, including by race and occupation [16].

Our study has some limitations. First, our numerator and denominator data come from different sources. People may misreport their ZIP code at the vaccination site or may move to another ZIP code since receiving the vaccine. The population denominator data are estimated for ZIP codes based on aggregation of census tract-level ACS estimates and under the

assumption that the population for 2020 to 2021 was similar to the population for 2015 to 2019. Second, sampling error in the denominators results in uncertainty in coverage estimates for specific ZIP codes. Third, a small percentage of MA residents did not provide ZIP codes when they received their primary series or booster vaccine. Fourth, we excluded 167 ZIP codes because they were nonresidential or institutional ZIP codes, including PO boxes, universities, and businesses with dedicated ZIP codes. Exclusion of these ZIP codes yielded a data set with greater comparability across units covering 97% of the MA population; however, some MA residents were not represented in our analysis. Due to these exclusions—particularly of university-specific ZIP codes—our totals for vaccination and booster coverage differ somewhat from published statewide estimates. Fifth, we further excluded the smallest ZIP codes ($n$ = 63) representing 1% of the population as the estimates were unstable due to very small denominators. Sixth, we lacked individual-level data that would enable inferences on the experiences of essential workers, individuals with different income and education levels, and of different race/ethnicities. Our inferences are therefore restricted to population-level statements about people in ZIP codes with different characteristics. Seventh, we do not distinguish between essential workers who were in healthcare and typically required to be vaccinated and in other sectors. Eighth, in assessing ZIP code racial/ethnic composition, we used a single metric for percent Black, Latino, or Indigenous, as these populations are concentrated in a relatively small number of MA ZIP codes. Our analysis misses important differences in the experiences of Black, Latino, and Indigenous MA residents. Ninth, it is possible that vaccine or booster coverage could be underestimated due to failure to link vaccine records longitudinally for individual residents; although all vaccinations are mandated to be reported into the Massachusetts Immunization Information System (MIIS), the foundation for the data used in our analysis, some data may be missing or incomplete, especially for those who may have received a shot outside of the state. Tenth, MA is one of the wealthiest states in the country, and patterns observed in MA may not hold elsewhere in the US.

These limitations should be considered alongside the study's strengths, namely, the use of official, state-reported data on primary series vaccine and booster shots that was aggregated from data on place of residence collected at vaccination facilities; *de novo* construction of population denominators at the postal ZIP code level enabling analysis of small-area variation in vaccine coverage not previously reported statewide; assessment of inequities across a range of ZIP code characteristics relevant to the epidemiology of COVID-19 and vaccine uptake; and assessment of inequities stratified by age. Finally, our analysis includes all persons vaccinated through October 10, 2022, about 18 months into the general population vaccine rollout.

Closing vaccine coverage gaps in MA and in other states will require ongoing concerted effort. The initial MA vaccination campaign in Spring 2021 was accompanied by daily media coverage and an explicit focus on equity. The late 2021 Delta and Omicron waves refocused public attention on the importance of completing the primary vaccine series and getting a booster shot. Once the 2021/2022 Omicron wave ended, vaccinations slowed. From March to September 2022, coverage of the primary series vaccine increased from 80% to 83% and booster coverage increased from 43% to 50%. A large share of MA residents have yet to receive the additional protection afforded by booster shots. Bivalent boosters were approved in September 2022, providing an opportunity to implement strategies in the near term to address inequitable access and coverage.

Our findings show where MA should focus its efforts: ensuring high, equitable booster coverage including among middle-aged and older adults, and vaccinating children and younger adults. Recent data indicate that vaccines for children are effective in reducing hospitalizations

[34]. Strategies used successfully during the initial vaccine rollout, including community outreach efforts and establishment of convenient venues for vaccination, need to be continued during the rollout of boosters to close coverage gaps in lower-educated communities regardless of racial composition.

In conclusion, our analyses indicate large geographic and sociodemographic inequities in vaccine and booster coverage, which should allow for targeted outreach efforts that leverage local infrastructure, including school-based immunization, workplace vaccination drives, community-based campaigns in multiple languages, and routine clinical care. In particular, the relatively lower uptake of booster shots (which is seen across the US) coupled with large disparities in who has received boosters indicates that increasing bivalent booster coverage should be a high-priority effort to protect against severe outcomes. Low booster coverage will lead to higher hospitalization and mortality rates in the future, with associated need for enhanced public health measures. Ensuring access and communicating the ongoing importance of vaccination and boosters will be essential.

## Supporting information

**S1 STROBE Checklist. STROBE Checklist.**
(PDF)

**S1 Fig. Percent (A) vaccinated and (B) boosted in urban areas.**
**Note**: S1 Fig displays higher-magnification maps of urban areas from the maps shown in Fig 1. To facilitate comparisons across ZIP codes within each panel (a) and (b), the scales in the panels differ. Shape files were obtained from the US Census Bureau, accessed October 20, 2022: https://www.census.gov/geographies/mapping-files/time-series/geo/tiger-geodatabase-file. html. Copyright protection is not available for any work of the United States Government (Title 17 U.S.C., Section 105).
(PDF)

**S1 Table. Age-adjusted bivariate associations of ZIP code characteristics with percent vaccinated and boosted.**
**Note**: Each cell shows results of a separate OLS (ordinary least squares) regression model. The crude models are bivariate. The age-adjusted include controls for ZIP code age shares in the following groups: 5–19, 20–39, 40–64, and 65+ years. Heteroskedasticity-robust 95% confidence intervals are shown in parentheses. $^*p < 0.05$, $^{**}p < 0.01$, $^{***}p < 0.001$.
(PDF)

**S2 Table. Sex-stratified models: Associations of ZIP code characteristics with percentage vaccinated and boosted.**
**Note**: Each column shows results of a separate OLS (ordinary least squares) regression model. The models are stratified by sex. Each includes controls for ZIP code age shares in the following groups: 5–19, 20–39, 40–64, and 65+ years. Heteroskedasticity-robust 95% confidence intervals are shown in parentheses. $^*p < 0.05$, $^{**}p < 0.01$, $^{***}p < 0.001$.
(PDF)

## Author Contributions

**Conceptualization:** Jacob Bor, Julia Raifman, Jonathan I. Levy.

**Data curation:** Jacob Bor, Kevin Lane.

**Formal analysis:** Jacob Bor.

**Investigation:** Jacob Bor.

**Methodology:** Jacob Bor, Kevin Lane, Jonathan I. Levy.

**Project administration:** Jacob Bor.

**Supervision:** Jacob Bor.

**Visualization:** Kevin Lane.

**Writing – original draft:** Jacob Bor, Sabrina A. Assoumou, Jonathan I. Levy.

**Writing – review & editing:** Jacob Bor, Sabrina A. Assoumou, Kevin Lane, Yareliz Diaz, Bisola O. Ojikutu, Julia Raifman, Jonathan I. Levy.

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
