## [Editor Report · Decision Letter 0]

18 May 2022

Dear Dr Bor, 

Thank you for submitting your manuscript entitled "Inequities in COVID-19 vaccine and booster coverage across Massachusetts ZIP codes: a population-based study following the Winter 2022 Omicron wave" for consideration by PLOS Medicine.

Your manuscript has now been evaluated by the PLOS Medicine editorial staff and I am writing to let you know that we would like to send your submission out for external assessment.

However, we first need you to complete your submission by providing the metadata that are required for full assessment. To this end, please login to Editorial Manager where you will find the paper in the 'Submissions Needing Revisions' folder on your homepage. Please click 'Revise Submission' from the Action Links and complete all additional questions in the submission questionnaire.

Please re-submit your manuscript within two working days, i.e. by May 20 2022 11:59PM.

Once your full submission is complete, your paper will undergo a series of checks in preparation for assessment. 

Kind regards,

Richard Turner, PhD

rturner@plos.org

---

## [Decision Letter · Decision Letter 1]

20 Jul 2022

Dear Dr. Bor,

Thank you very much for submitting your manuscript "Inequities in COVID-19 vaccine and booster coverage across Massachusetts ZIP codes: a population-based study following the Winter 2022 Omicron wave" (PMEDICINE-D-22-01626R1) for consideration at PLOS Medicine. 

Your paper was evaluated by a senior editor and discussed among all the editors here. It was also sent to independent reviewers, including a statistical reviewer. The reviews are appended at the bottom of this email and any accompanying reviewer attachments can be seen via the link below:

[LINK]

In light of these reviews, I am afraid that we will not be able to accept the manuscript for publication in the journal in its current form, but we would like to consider a revised version that addresses the reviewers' and editors' comments. Obviously we cannot make any decision about publication until we have seen the revised manuscript and your response, and we plan to seek re-review by one or more of the reviewers. 

We expect to receive your revised manuscript by Aug 10 2022 11:59PM. Please email us (plosmedicine@plos.org) if you have any questions or concerns.

We look forward to receiving your revised manuscript. 

Sincerely,

Beryne Odeny, 

PLOS Medicine

plosmedicine.org

1) Please revise your title according to PLOS Medicine's style. Your title must be nondeclarative and not a question. It should begin with main concept if possible. Please place the study design in the subtitle (i.e., after a colon), e.g., a retrospective cohort study. Please place the statement, “following the winter 2022 Omicron wave” before the colon.

2) Please include line numbers in your next draft.

3) Abstract:

a) Please ensure that all numbers presented in the abstract are present and identical to numbers presented in the main manuscript text.

b) Please mention the study design at the start of the Methods and Findings section

c) Please quantify the main results (please present both 95% CIs and p values).

d) In the last sentence of the Abstract Methods and Findings section, please describe the main limitation(s) of the study's methodology.

e) Conclusions: 

i) Please interpret the study based on the results presented in the abstract, emphasizing what is new without overstating your conclusions. 

ii) Please mention specific implications substantiated by the results.

4) Author Summary – under the subheading, “What did the researchers do and find?” please trim the text to no more than 4 bullet points

5) Please conclude the “Introduction” with a clear description of the study question or hypothesis. 

6) Did your study have a prospective protocol or analysis plan? Please state this (either way) early in the Methods section. 

7) Please provide the name(s) of the institutional review board(s) that provided ethical exemption.

8) Please ensure that the study is reported according to the STROBE and include the completed STROBE checklist as Supporting Information. Please add the following statement, or similar, to the Methods: "This study is reported as per the Strengthening the Reporting of Observational Studies in Epidemiology (STROBE) guideline (S1 Checklist)."

9) How was race/ethnicity defined and by whom?

10) Please provide p values (where appropriate) in addition to 95% CIs in the main text and tables

11) Figures and tables:

a) Please define all abbreviations in Tables and Figures e.g., SD, HH, Blk, Hisp, Indig, OLS

b) Please confirm that the appropriate usage rights apply to the use of this map. Please see our guidelines for map images: https://journals.plos.org/plosmedicine/s/figures#loc-maps

c) Please consider avoiding the use of red and green in order to make your figure more accessible to those with color blindness

12) References: 

a) Please select the PLOS Medicine reference style in your citation manager. In-text reference call outs should be presented as follows noting the absence of spaces within the square brackets, “…population [1,2].”

b) References should have six names before et al. 

c) Please provide a weblink for Reference #22

d) Please ensure all weblinks are accessible

e) Please ensure that journal name abbreviations consistently match those found in the National Center for Biotechnology Information (NCBI) databases. https://journals.plos.org/plosmedicine/s/submission-guidelines#loc-references. 

Comments from the reviewers:

Reviewer #1: I confine my remarks to statistical aspects of this paper. These were well done and I recommend publication.

Peter Flom

Reviewer #2: Overall comments: 

This is a well-written manuscript examining inequity in vaccination and boosters in Massachusetts using aggregated state-level data. The authors use sophisticated methods to grapple with some of the discrepancies between the state-level vaccination data and population census data. The authors' present interesting and new data on associations between zip code level characteristics and vaccination and booster rates; it is an important finding to see how these disparities differ for initial vaccination and boosters. Most of the comments focus on areas that could be strengthened or clarified a bit. Mostly, I think the framing of the fact that associations reverse with adjustment could be sharper…I think having this crystal clear is important so readers do not take away the "wrong" message from it. I think additional exploration of the VEI program would be quite interesting…any insights into what potential impact it may have had I think would be very important to know (acknowledging there may be many limitations to being to do such an analysis).

Specific comments:

Intro

* The introduction is well-written. I think the systemic nature of root causes of the disparities could be spelled out a bit more. Many of the mechanisms of the systemic racism in healthcare are there, but they could be tied together. Also could bring in the fact that there are different phenotypes of vaccine hesitancy/confidence, which is also rooted in disparities. 

Methods:

* I wasn't fully clear on the imputation approach for the neither male or female. Did you use the ratio for the zip codes where this information was available (there were a few zip codes with this information) to impute all the rest? Where there instances where the imputation estimated greater than 30 people for a zip code (in which case you would expect that data should not have been missing…perhaps needs a cap too). Is neither male nor female available in census data to understand the ratio in the total population.

* Also, do estimates change at all if you used age (I imagine this data is also fairly complete?). I think another approach that could be considered is multiple imputation that could potentially incorporate all the reported vaccine data and zip code level characteristics. Hopefully all of these would generate fairly similar results, but some sensitivity analyses to confirm might be nice.

* For the zip by age analyses, it is possible to do a similar imputation process to not exclude those with missing cells. This might require a more complicated imputation process since age distributions will likely vary across zip code level characteristics too. Just unclear what, if any, impact this missingness has on results. Might also be helpful to present this missingness information so that extent of it is transparent.

* The methods used to estimate percent vaccinated or boosted that accounts for the uncertainty in population estimates is sophisticated. However, I wasn't clear where the 90% confidence intervals for percent vaccinated came into play for any of the analyses. Figures 3 to 5 seem to be using something else, but I may be mistaken.

* I might separate out the different analyses under zip code-level and zip code by age (each its own paragraph). Particulalry for the regression and interpretation of adjustment, there is enough to warrant separating it out. I had a bit of a hard time matching them up with figures an results as I went through ti.

Results

* I wonder if by the geospatial plots it would be helpful to make the scale for vaccinated and boosted plot the same. Currently they seem to be color-coded by decile, but the most prominent finding then becomes that zip codes in the top/bottom deciles for the initial vaccine series remain there for boosters as well. Perhaps that is the point trying to be conveyed though. Either way, a figure legend that has a bit of walkthrough to highlight the main findings would be helpful.

* Figure 2 is quite informative. It shows clear relationships between zip code characteristics and vaccination (and stronger relationship for boosters). Reporting out the confidence intervals around the estimates would nice and also the p-value or correlation coeffiecient (I see that it is in Table 2 and so could also just make clear the estimates are there). 

* The paragraph starting with "Bivariate association…" seems like it should be included in the methods section to explain the rationale for the different regression models.

* I found Table 2 kind of hard to interpret. The reversal of the slope with adjustments seems a bit unexpected. Is there concern for overfitting (not sure there would be with 418 zip code)? The authors do explain this in the results and discussion, though it is still hard to align with other data in the literature (which often show independent associations of all these factors). It may be that the in Massachusetts, the effects of systemic racism are almost fully mediated through income and education (as opposed to many other areas of the countries where this is not the case). 

* I think something else worth considering is that it may also be reflective of success of the VEI program in that it was able to attenuate many of these associations (or reverse them). In the final model, being a VEI program was associated with only 0.5% difference in vaccination…which seems very low. Though not able to assess here, I would expect that difference to likely be higher. 

* Ultimately, I might try to frame some of the discussion on causal interpretation thinking about mediators of racism and disparities. I always find Camara Jones work quite informative in interpreting these types of analyses/findings (https://academic.oup.com/aje/article/154/4/299/61900). There are complex relationships between income, education, race, employment, and zip code and it is not as straightforward to be able to just adjust them away. Careful interpretation is required (which I think that authors do) but perhaps could go a bit deeper.

* The impact of age structure is actually quite interesting and something I think that hasn't always been taken into account when considering raw estimates. It might also be interesting to see how this impacts the bivariate estimates too. 

* I think a bivariate estimate for VEI could also be interesting. To accomodate all of the different estimates, could consider presenting all bivariate estimates in a single column.

* Consider providing details in table 2 about coefficient interpretation (i.e., X% change per 1% increase…could also scale similarly to Figure 2).

* Is it possible to include case and death rates as zip code level covariates too? Not sure how readily available that data is, but is quite relevant to the targeting that happened by the VEI program it would seem.

* For Figure 3, I do know that it common enough that vaccination rates seem to exceed population estimates, though it is still odd to see. Most of the approaches to exclusion or missing data would seem to underreport vaccination rates. I am not sure if there is any strategy to address it, but could consider truncating at 100%. Would address the approaches to this in the methods.

* For Figure 4, consider putting vaccinated and boosted on same panel (and color code differently) and aligning the a-d on one page. Would also add figure legend that explains lowest to highest decile for quicker orientation for the reader. Also consider putting children on the same figure (as opposed to figure 5).

* The authors assess age strata by zip code characteristic interactions. Is it also possible to assess race by zip code characteristic interactions also. This data seems available, though missingness may be a problem. It doesn't seem possible to essentially analyze at individual level using frequency weights, but this could be included as a limitation.

* Are the authors' aware of changes to the VEI efforts during the initial campaign vs. for boosters. I imagine there were. If the VEI initiative was more prominent during the initial vaccination campaign, could explore the interaction of its effect during these two periods. This is already alluded to in results for vaccination vs. booster but could be more prominent and formally assessed. Should also be included in the discussion.

* 

Discussion

* I think something that ends up somewhat de-emphasized in the discussion how associations changed after adjustments is that being a minority in the US and experiencing systemic racism is can't really be disaggregated from income, education, where you live, and healthcare access. The ultimate conclusions the authors reach are fine, but I think the meaning of why some of these associations attenuated after adjustment needs to be sharpened. Again Camara Jones' work is illustrative here. What comes across a bit—and I don't think the authors intended this—is that race, employment aren't the root causes and it is just income and education.

* I think this finding probably needs to be contextualized in the setting of Massachusetts (I would expect the findings would differ in many other parts of the US) and perhaps also in the context of the VEI program. I am not sure it would be too far fetched to hypothesize that VEI program may be part of the reason zip codes with more minorities and essential workers outperformed expectations after adjustment specifically they were targeted for additional resources. The authors can consider whether this is reasonable or not.

* I think additional limitations to consider are that 1) this analysis is primarily ecological and could not use individual-level data, 2) missing data, and 3) also the discrepancy in denominators for vaccination and the census data (this is alluded to already but could be more specific).

* I think in general more contextualization is needed in terms of what the findings mean beyond the different analytic strategies (e.g., this analysis was adjusted this one was not). What is larger of importance of what the results mean because of these analytic differences.

* For example, I think more contextualization is needed in terms of this finding "In fact, the higher risk of severe COVID-19 in low-income households seems to be strongly driven by personal occupational status and a foreign background." Why would this be the case? What are the mechanisms

Aaloke Mody

Reviewer #3: This is an excellent, well-written article on disparities in COVID-19 and vaccine booster uptake in Massachusetts. The authors take a careful look at racial/ethnic and other group differences in vaccine/booster uptake, adjusting for age and other factors. Among their key and important findings are that many racial/ethnic differences that were assumed to be due to differences in vaccine hesitancy are likely instead attributable to differences in age and SES distributions. The authors' framing of the issues is excellent. The methodological approaches are thoughtful and rigorously applied. The data are very timely (through March 2022). Both primary vaccine and booster uptake are examined. 

I have only a few minor suggestions that the authors may want to consider:

1. There have been some concerns about the accuracy of vaccine coverage data from state and local health department vaccine registries. The authors mention many of the reasons why there are some limitations to the data. However, one that they don't mention is that accurately estimating vaccine coverage requires linkage of longitudinal information across individuals (i.e., dose 1, dose 2, and booster doses). If there is not good matching, this can result in underestimating or overestimating coverage. When fully vaccinated person A's dose 1 and dose 2 are not linked, it could be counted as two individuals with single doses of vaccine only. When person B and person C's first doses are incorrectly linked, then one person will be counted as fully vaccinated and not two as under vaccinated. etc. I am not sure how much of an issue this is for Massachusetts, but the authors may have some information worth sharing in the article about the data quality of the numerator. In other jurisdictions and nationally, there have been some implausible estimates of coverage by race/ethnicity, particularly on the higher end. But they may want to say more about the data generating mechanisms and data sources for the numerator data.

2. Could the authors clarify the source of ZIP level occupation data?

3. The authors mention that a key aim of the regression analysis is to determine whether bivariate associations are explained by other factors. Would it be helpful to include DAGs in the appendix, to differentiate potential confounding vs potential mediating? Or at least to discuss some of the possible reasons why bivariate associations may be fully or partially or not at all explained by other factors?

4. While not possible to estimate at the ZIP code level without joint distributions, it may be important for the authors to recognize the potential role of prior history of SARS-CoV-2 infection as a possible factor weighing on people's decisions to pursue a vaccine or booster. i.e., those who had covid in the past, including recent past, may be less likely to pursue a booster. And also that in considering at least the short term risks for severe outcomes during surges, it would be helpful to know the extent to which populations (say ZIP codes or by SES strata) the proportion of the population that has protection due to vaccination/boosting only or vaccination/boosting PLUS a recent history of COVID. This may also explain some of the differences between groups, including essential workers (and other groups who have endured a disproportionate share of the pandemic's burden) vs. others.

Reviewer #4: Thank you for the opportunity to review this research.

This is a well written paper.

1. For the multivariable regression（Table 2a and Table 2b), how was age distribution adjusted? Do it mean age, as a continuous variable, was included in the models?

2. Figure 1: what's the difference between "missing" and "suppressed". How many zip codes were excluded because of missing ness, and how many were excluded because they corresponded to specific universities or businesses, they did not correspond to residential areas, or they were assigned to post office boxes and not to residential addresses, o

3. Figure 3: why the point estimate and upper bound of the confidence exceed 1? Is there any method that could make the simulated estimate less than 1?

4. Through stratified analysis, it's interesting that larger inequities in vaccination rates among younger adults and children were found. The author visually presented the the difference in the inequities in vaccination rates across different age groups (Figure 4), but whether the differences are statistically significant have not been tested (Table 3).

5. In addition to age, could sex be a potential effect modifier as well?

[LINK]

---

## [Decision Letter · Decision Letter 2]

12 Dec 2022

Dear Dr. Bor,

Thank you very much for re-submitting your manuscript "Inequities in COVID-19 vaccine and booster coverage across Massachusetts ZIP codes in Fall 2022: a population-based cross-sectional study" (PMEDICINE-D-22-01626R2) for review by PLOS Medicine.

I have discussed the paper with my colleagues and the academic editor and it was also seen again by 3 reviewers. I am pleased to say that provided the remaining editorial and production issues are dealt with we are planning to accept the paper for publication in the journal.

[LINK]

We look forward to receiving the revised manuscript by Dec 19 2022 11:59PM.   

Sincerely,

Philippa Dodd MBBS MRCP PhD

PLOS Medicine

plosmedicine.org

Requests from Editors:

GENERAL

Please respond to all editor and reviewer comments detailed below, in full

Line numbers should start at 1 on the title page and continue in sequence to the end of the manuscript. In your response to revisions please indicate where in the manuscript the revision can be located (by section and line number)

DATA AVAILABILITY STATEMENT

Thank you for making your data available. Your data availability statement requires revision. Please also provide the URL for the Massachusetts Department of Public Health and US Census Bureau, in the manuscript submission form.

TITLE

Thank you for revising your title. Previously you detailed “following the Winter 2022 Omicron wave” now you detail “across Massachusetts ZIP codes in Fall 2022”. In your abstract you state “vaccination and booster coverage 18 months into the general population vaccine rollout…as of October 2022”. The different time descriptions used in the two versions of your title describe different time periods (suspect that in the former you mean Winter 2021 Omicron wave?). 

Does the time frame need to be defined by season? Also, the term “fall” is not universally adopted to describe the Autumn season. Suggest instead: “Inequities in COVID-19 vaccine and booster coverage across Massachusetts ZIP codes after the emergence of Omicron: a population-based cross-sectional study” or : “Inequities in COVID-19 vaccine and booster coverage across Massachusetts ZIP codes in October 2022: a population-based cross-sectional study” or a similar variation of either. 

ABSTRACT

Line 19-20: would it be helpful to also report the data that you refer to in this statement?

AUTHOR SUMMARY

Please remove the box and shading surrounding the author summary – it should be included as text rather than in figure form (and should be included in the line numbering also)

“Percent college graduates was the strongest predictor of vaccine and booster uptake.” Would it be helpful to simply state “education level” or something similar?

“These impacts are preventable.” Suggest tempering of language here, especially as you state “inequities…may lead to…”. Suggest “Inequities in vaccine and booster coverage may lead to preventable inequities in morbidity, mortality, and economic losses due to COVID-19.” Or something similar.

INTRODUCTION

Page 4, line 1: Please change the heading “Background” to “Introduction”

Page 5, line 2: Please define MA at first use here as follows: “Massachusetts (MA)”

METHODS and RESULTS

Page 7 lines 29-30: suggest the following “….across ZIP codes: percentage of ZIP code residents aged 0-4 years, 5-19 years, 20-39 years…” Note the removal of the repeated % symbol. This also makes your reporting consistent with that on page 6, line 34. Please check and amend throughout all sections of the manuscript

Page 10 paragraph 1 – it would be helpful to enumerate the percentages reported here

Page 12, line 40: “40 CI -7.3 to -0.1, p<.05” please report p as <0.05 – please check and amend throughout including tables and figures as necessary

Figure 3: when describing figure 3 you refer to “columns” but they are not columns they are graphs stacked upon each other. Please refer to plots or graphs “(left sided graphs)” for example

*** Regarding the reviewer’s comment re: sex as a potential modifier. Any changes to analyses that are peer review driven should and can be included in the manuscript as part of PLOS Medicine’s transparency of data reporting policy. Changes in the analysis should be identified as such in the Methods section of the paper, with rationale. Outcome of the analysis should be reported and/or discussed io the relevant section of the manuscript. The data table can be included in the supporting files. Please include a caption which describes its content. As for all tables, please report p-values (not only asterisks), 95% CIs, define all abbreviations, report whether any other variables are adjusted for, report unadjusted analyses for comparison. ***

FIGURES and TABLES

All figures and tables should have an appropriate figure caption such that the contents of the figure/table are accessible without the manuscript text. Please include as required including the supplementary files.

Figure 2 Title: suggest “Percent vaccinated and boosted according to individual ZIP code characteristics” or something similar

Figure 4: Please use full term “indigenous” the abbreviation could be considered offensive. Without the manuscript text to refer to, it is not easy to understand what the figure shows. What does "ZIP code decile" mean? Please revise in-line with above comments. 

TABLES

Table 1: Considering the study shows disparity between ZIP codes, is it necessary to show combined data in a table here? If so, suggest revising the title to “Mean Vaccination Coverage (percentage) Across the State of Massachusetts According to Age” or something similar. 

We ask that all authors provide a table showing the baseline characteristics of the study population and specific to your study, ZIP code demographics should also be included. Perhaps the currently presented data referred to above, could be incorporated into this table.

Table 2: Thank you for reporting p-values. We request that these are reported in full in a separate column, rather than defined by asterisks. Please revise accordingly, this may require you to split your tables. For main outcome measures, where analyses are adjusted, we request of all authors that the unadjusted analyses also be reported. Please indicate which factors are adjusted for in the table caption.

We note reviewer comments re: presentation of Table 2 and agree that accessibility could be improved, the empty cells in the tables are rather distracting and in light of the aforementioned required revisions, we request that the table is revised accordingly.

Table 3: As above. Please report the p-value result in a separate column in place of asterisks. Please provide unadjusted analyses as necessary.in the table caption please indicate which factors are adjusted for.

Please define VEI in the table caption

DISCUSSION

Please ensure that the Discussion is organized and presented as follows: a short, clear summary of the article's findings; what the study adds to existing research and where and why the results may differ from previous research; strengths and limitations of the study; implications and next steps for research, clinical practice, and/or public policy; one-paragraph conclusion.

REFERENCES

Page 4, line 15-16: In-text reference callouts should be placed in square parentheses and preceding punctuation as follows: “…please check [1,2,3,4].” Please note the space after the text before the opening parenthesis. Please check and revise throughout the manuscript.

In your bibliography please ensure you list up to but no more than 6 authors (followed by et al where more than 6 authors contribute to the cited study

Journal name abbreviations should be those found in the National Center for Biotechnology Information (NCBI) databases. 

Please see out guidelines for further information here: https://journals.plos.org/plosmedicine/s/submission-guidelines#loc-references

SOCIAL MEDIA

To help us extend the reach of your research, please provide any Twitter handle(s) that would be appropriate to tag, including your own, your coauthors’, your institution, funder, or lab. Please respond to this email with any handles you wish to be included when we tweet this paper.

COMMENTS FROM THE ACADEMIC EDITOR

Identifying potential areas of inequity for COVID-19 vaccines is certainly an important area. I read through the author responses and the article. It has done a nice job using available data to provide some insights for MA. Would agree with the reviewers and accept at this time.

Comments from Reviewers:

Reviewer #2: The authors were very responsive to reviewer comments. I have no further suggestions.

Reviewer #3: The authors have been very responsive to comments from reviewers. It is an excellent paper!

Reviewer #4: My questions have been sufficiently answered. I do not have any further comments. Thnak you.

[LINK]

---

## [Editor Report · Decision Letter 3]

28 Dec 2022

Dear Dr Bor, 

On behalf of my colleagues and the Academic Editor, Dr. Amitabh Suthar, I am pleased to inform you that we have agreed to publish your manuscript "Inequities in COVID-19 vaccine and booster coverage across Massachusetts ZIP codes after the emergence of Omicron: a population-based cross-sectional study" (PMEDICINE-D-22-01626R3) in PLOS Medicine.

Prior to publication please address the following final revision:

* Page 10, line 376 onwards: regarding descriptive information and percentages, please also include numerators (and denominators) used to derive these – i.e. what number of people aged 65+ did the 100% coverage equate to? Apologies for my previous lack of clarity regarding this point.

PRESS

Best wishes, 

Philippa Dodd, MBBS MRCP PhD 

PLOS Medicine